

# Precision measurement of the index of refraction of deep glacial ice at radio frequencies at Summit Station, Greenland

Christoph Welling[1] and The RNO-G Collaboration [*]

[1]Dept. of Physics, Enrico Fermi Inst., Kavli Inst. for Cosmological Physics, University of Chicago, Chicago, IL 60637, USA
[*]A full list of authors appears at the end of the paper.

**Correspondence:** Christoph Welling (christophwelling@uchicago.edu, authors@rno-g.org)

**Abstract.** Glacial ice is used as a target material for the detection of ultra-high energy neutrinos, by measuring the radio signals that are emitted when those neutrinos interact in the ice. Thanks to the large attenuation length at radio frequencies, these signals can be detected over distances of several kilometers. One experiment taking advantage of this is the Radio Neutrino Observatory Greenland (RNO-G), currently under construction at Summit Station, near the apex of the Greenland ice sheet. These experiments require a thorough understanding of the dielectric properties of ice at radio frequencies. Towards this goal, calibration campaigns have been undertaken at Summit, during which we recorded radio reflections off internal layers in the ice sheet. Using data from the nearby GISP2 and GRIP ice cores, we show that these reflectors can be associated with features in the ice conductivity profiles; we use this connection to determine the index of refraction of the bulk ice as $n = 1.778 \pm 0.006$.

## 1 Introduction

Cosmic rays have been of interest to physicists for over a hundred years, but the sources of the most energetic, so-called ultra-high energy cosmic rays (UHECRs) have so far evaded discovery. This is partly due to their charge, which causes a cosmic ray to be deflected by magnetic fields during its propagation and obscures the direction of its source. A solution to this problem would be the detection of ultra-high energy neutrinos, which are expected to be produced by the same sources, as well as by UHECRs during propagation. Since they are electrically neutral, neutrinos are not affected by magnetic fields and can propagate through most obstacles, which might block other messengers. However, this means a neutrino is also likely to pass through any detector one may build. At EeV scale energies, this, along with the low expected neutrino flux, necessitates the construction of giant detectors with volumes on the scale of $100 \mathrm{km}^3$ or more.

The Radio Neutrino Observatory Greenland (RNO-G) (Aguilar et al., 2021), and other similar radio neutrino detectors (Hoffman, 2022; Anker et al., 2019; Aartsen et al., 2021) are based on the detection of radio signals by antennas embedded in glacial ice. If a neutrino interacts in the ice, it produces a particle shower, which emits a short radio pulse via the Askaryan effect (Askaryan, 1961; Alvarez-Muniz et al., 2011). Thanks to the large attenuation length of cold ice at radio frequencies (Aguilar et al., 2022a, c; Avva et al., 2015; Barrella et al., 2011; Hanson et al., 2015), a shower produced by a neutrino at energies above $\sim 10 \mathrm{PeV}$ can be observed over distances of several kilometers, making it possible to monitor several cubic kilometers with a small number of antennas, and achieve an effective volume large enough for UHE neutrinos. If the radio





signal from a neutrino is detected, it is also possible to reconstruct the neutrino energy and direction from the radio signal
(Aguilar et al., 2022b; Plaisier et al., 2023).

In order to use ice as a detection medium, a thorough knowledge of its dielectric properties at radio frequencies is necessary.
To this end, a series of calibration campaigns has been undertaken at Summit Station, where RNO-G is located, and will
continue in coming years. These included measurements of the ice attenuation length using the backscatter of radio signals off
the bedrock (Aguilar et al., 2022a, c). In addition to the bedrock echo, reflections were also observed from within the ice sheet.
Though the reflectivity of these layers is rather low, a shallow reflector is a potential source of background for a radio neutrino
detector: if an air shower impacts on the ice surface, it produces a radio signal. If this signal is reflected upwards, it is hard to
distinguish from a radio signal that originated in the ice, like that from a neutrino (De Kockere et al., 2022).

On the other hand, these layers may present an opportunity to study the dielectric properties of the ice. Radio reflectors in
deep ice result from some dielectric contrast, perhaps caused by rapid changes in the ice conductivity; this connection has been
demonstrated qualitatively at the site of the Greenland Ice Core Project (GRIP), about 27km from Summit Station (Hempel
et al., 2000). This was done by first determining the wave velocity of the bulk ice from the change in the travel time of the
return echo from an in-ice reflector, as the distance between transmitter and receiver antenna is varied (common depth point
method). The known wave velocity was then used to determine the depth corresponding to a specific reflection. Our goal in this
paper is to turn this around: By leaving the velocity as a free parameter, and utilizing the Greenland Ice Sheet Project 2 (GISP2)
ice core at Summit Station, as well as the GRIP core, we use the association between radio reflections and ice conductivity
to measure the index of refraction of the bulk ice with sub-percent precision. A similar approach has been used to determine
the ice permittivity near Dome C in Antarctica (Winter et al., 2017), but no uncertainties were given, as this was not the main
goal of that publication. Furthermore, permittivity measurements of ice in the lab are also available Johari (2008); Bohleber
et al. (2012). To our knowledge, this paper is the most precise in situ measurement of the index of refraction of deep ice in
Greenland thus far.

The index of refraction is an important property in radioglaciological surveys, which are often used to measure ice depths
and identify points of interest for further glaciological studies (Plewes and Hubbard, 2001), in order to calculate the depth from
the return time of the radio signal.

For a radio neutrino detector like RNO-G, the index of refraction is of particular interest because of the role it plays for the
radio signal emission via the Askaryan effect, as well as its propagation from the shower to the detector. The radio signal is
emitted around a cone, with an opening angle given by the Cherenkov angle

$$\theta_C = \arccos\left(\frac{1}{n}\right) \tag{1}$$

which depends directly on the index of refraction $n$ of the ice trough which the shower propagates. To reconstruct the direction
of a neutrino detected by RNO-G, the *viewing angle*, i.e. the angle between the neutrino direction and the direction into which
the radio signal is emitted, has to be measured. This can be done using the spectrum of the radio signal, which depends on the
difference between the *viewing angle* and the Cherenkov angle. Thus, an error on the Cherenkov angle will directly lead to an
equal error on the *viewing angle* and the neutrino direction. A $1\%$ error in $n$ corresponds to a $0.4°$ error in the Cherenkov angle,





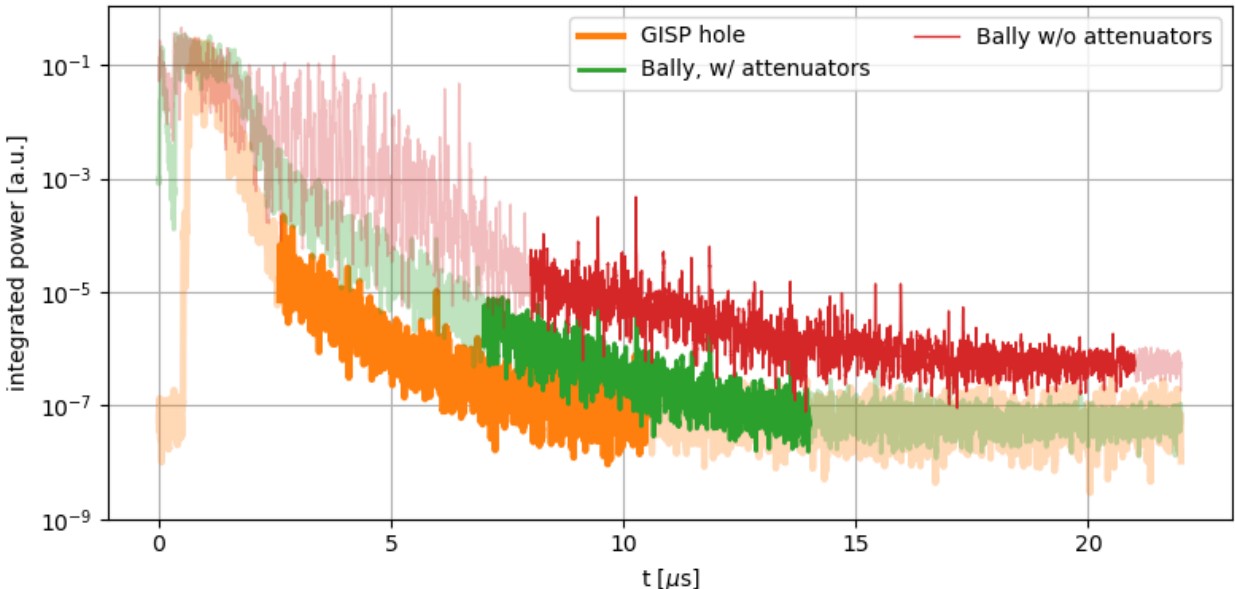

**Figure 1.** Return power of the radio signal from three measurements taking at Summit Station in 2022. Transparent sections mark the times when the signal was affected by amplifier saturation or is dominated by noise.

which, at first glance, seems small compared to the several degrees angular resolution of RNO-G (Plaisier et al., 2021, 2023).
However, this comparison obscures the fact that the uncertainty on the direction of a neutrino detected by RNO-G is highly asymmetric. On a sky map, the error contour resembles an ellipse whose two axes are defined by the uncertainty on the *viewing angle* and the much larger uncertainty on the polarization of the radio signal. The *viewing angle* resolution is $\sim 0.5°$, so a 1% error on $n$ would significantly increase the area of the error ellipse, and therefore the part of the sky that would have to be searched when doing multimessenger astronomy with RNO-G.

The bulk ice index of refraction also represents a boundary condition for the index of refraction profile in the firn, which is essential to reconstruct the position of the neutrino interaction (Aguilar et al., 2022b). The index of refraction profile of the firn also influences the effective volume of the detector, by creating a so-called *shadow zone*, where the propagation of the radio signal to the detector is suppressed (Barwick et al., 2018; Deaconu et al., 2018).

## 2   Radio Echo Measurements

The radio echo measurements used in this paper were carried out in the summer of 2022 at Summit Station, near the GISP2 borehole. They are a follow-up to measurements done in 2021 with the goal of measuring the radio attenuation of the ice (Aguilar et al., 2022a, c). The setup is similar to the previous one with the main change being the replacement of the log-periodic dipole antennas with horn antennas. The horn antennas have a smaller group delay, which produces a shorter radio





pulse and improves the timing resolution. It also reduces interference between return signals from proximal reflectors.

Signals were produced by an IDL-2 pulse generator and split into two outputs. One output was fed into a 145MHz highpass filter and then into one of the horn antennas through an MILDTL17 and an LMR240 coaxial cable. The other output, used as a trigger signal, was attenuated by 40 dB and fed into an oscilloscope via an MILDTL17, an LMR240 and an LMR400 cable. Both the transmitting and the receiving antenna were buried in the snow on opposing sides of the GISP2 hole, about 102 m

from each other. The receiving antenna was connected to an amplifier of the same type as used by the shallow component of an RNO-G station via a MILDTL17 coaxial cable, and connected from there to the oscilloscope with an LMR240 cable. Because the echo from a single radio pulse quickly falls below the noise background, 12000 waveforms were averaged and recorded at a sampling rate of 2.5 GHz. The strong air-to-air signal between the antennas caused the amplifier to saturate, requiring some time to recover, making the first 2.6 μs not usable, and the radio echo falls below the noise background around 10.6 μs after the

trigger, which sets the range of depths that are observable with this measurement.

Two additional measurements were taken about 550 m away from the GISP2 hole, in the vicinity of the so-called "Bally Building". Signals were produced by an AVTECH AVIR-1-C pulse generator, which provides more output power and a faster trigger rate, but could not be used at the GISP2 hole because of a lack of a suitable power source. This setup allowed to average 30000 waveforms and detect reflections from deeper in the ice, but also increased the time the amplifier needed to recover from

90 saturation. Therefore, another run was done with the same setup, but 12 dB of attenuation added to the signal chain on the transmitting antenna. This mitigated the amplifier saturation, but also caused the radio echo to fall below the noise floor sooner and increased the system noise figure.

From each run, we calculate the return power of the radio signal in a sliding rectangular time window with a width of 10 ns, corresponding to roughly one period of the radio signal at the lowest frequency in the band. The result is shown in Fig. 1.

The radio signal power is then corrected for the propagation distance using the attenuation length measured in (Aguilar et al., 2022a). [1]

Because of the distance from the GISP2 hole, layers may be at different depths under the Bally building, which could cause a significant, and difficult to estimate, systematic error on $n$. Therefore, we will only use the measurement directly at the GISP2 hole to measure the index of refraction. We nevertheless compare the measurements at the Bally building to the ice core data,

to demonstrate that the connection between ice conductivity and radio reflectors holds to greater depths and could be used to improve on this measurement in the future. To do so, we first determine the time offsets between the different measurements by correlating the attenuation-corrected return power as a function of time-since-trigger. Then all three measurements are combined into a single time series. In cases for which more than one measurement overlap, the average return power is used.

---

[1]To convert the arrival time of the radio pulse to the propagated distance, an assumption about the index of refraction $n$ and any time offsets $\Delta T$ due to e.g. cable delays already has to be made here. One could redo this correction for each value of $n$ and $\Delta T$, but this dramatically increases the computing demands. The choice of $n$ and $\Delta T$ at this stage turns out to have a negligible impact on the final result, so we ignore this complication here.



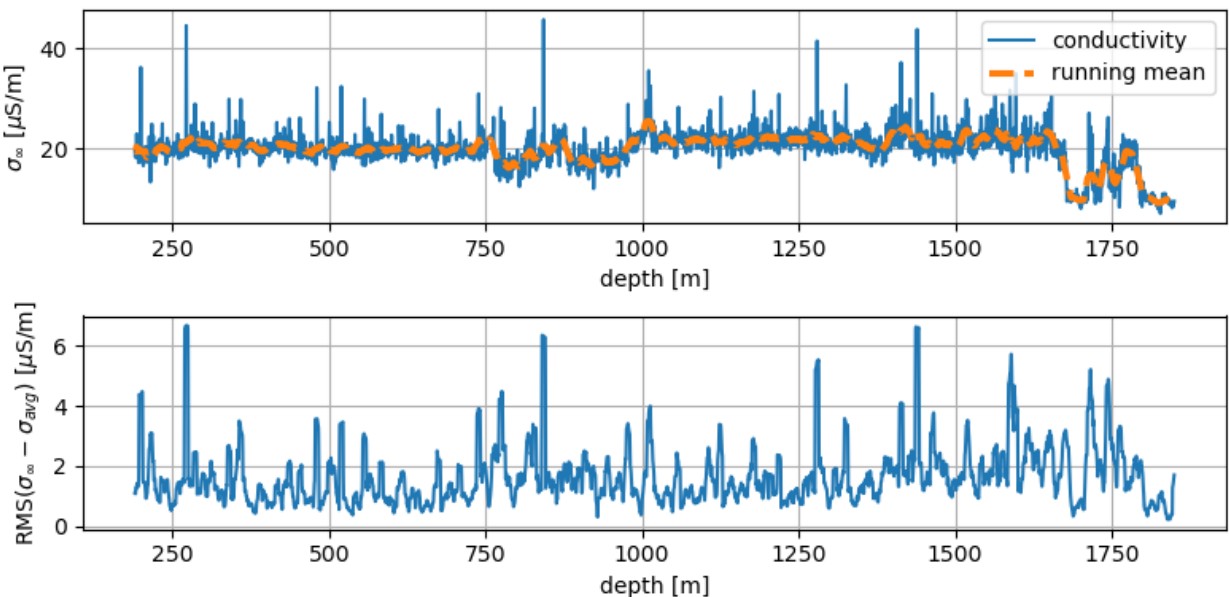

**Figure 2.** Top: AC conductivity data $\sigma_\infty$ from the GRIP ice core, adjusted to the corresponding depths at the GISP2 site, overlaid with the running mean of the conductivity. Bottom: Root mean square (RMS) of the deviation of the conductivity from the running mean.

## 3 Index of Refraction Measurement

The principle of the index of refraction measurement is rather simple. If we are able to match reflectors from the radio echo measurements to features in the conductivity data from the ice core, the index of refraction can be calculated as

$$n = \frac{c_0 \cdot \Delta t}{2 \cdot \Delta z} \tag{2}$$

where $c_0$ is the vacuum speed of light, $\Delta t$ is the time between observed radio echos and $\Delta z$ is the difference in depth of the reflectors. The direct current (DC) conductivity of the GISP2 ice core has been measured for its entire depth range (Taylor, 2003), but the relevant property governing the effect on radio waves is the alternating current (AC) conductivity $\sigma_\infty$, which has not been determined for GISP2. The AC conductivity has been measured using dielectric profiling (DEP) for the nearby Greenland Ice Core Project (GRIP) core (Greenland Ice Core Project, 1994; Wolff et al., 1995). Both ice cores were taken only 28 km apart, and the DC conductivity measurements are well correlated up to depths of 2700 m (Taylor et al., 1993). It is therefore reasonable to assume that the AC conductivity at GISP2 is similar to that at GRIP, though there are offsets between layer depths at the two sites, which we correct for based on (Rasmussen et al., 2014; Seierstad et al., 2014; Centre for Ice and Climate, Niels Bohr Institute, 2014).

For a given index of refraction, the signal return time $t$ can be converted to a reflector depth using

$$z_0 = \frac{1}{2} \cdot \frac{c_0}{n} \cdot (t - \Delta T) \tag{3}$$



if the distance between transmitting and receiving antenna is negligible. If they are further apart, as is the case here, the
additional travel distance is accounted for via the expression

$$z = \sqrt{z_0^2 - 0.25 \cdot d^2} \tag{4}$$

where $d$ is the distance between the transmitting and receiving antennas, 102 m in our case. The changing index of refraction
in the firn causes radio signals to be bent downward as they propagate, which causes a small change to the propagation time.
We simulated this effect using the analytic raytracing method Glaser et al. (2020) and found it to be negligible.

$\Delta T$ is a second free parameter which accounts for time offsets due to cable delays, a changing index of refraction in the
firn, and the unknown offset between the zero depth point of the GISP2 core and the location of the antennas. Our strategy to
measure the index of refraction is to vary $n$ and $\Delta T$, convert the radio signal return times to the corresponding depths, and
calculate the correlation between the ice conductivity at this depth and the return power. For depths between two conductivity
measurements, the value is linearly interpolated between the two closest data points.

However, we do not directly correlate the AC conductivity with the radio echo power. Radio reflections at large depths are
thought to be caused by rapid changes in the AC conductivity of the ice. Therefore, rate of change of $\sigma_\infty$ is more important
than the value of $\sigma_\infty$ itself. We therefore average the conductivity over a sliding window with a width of 5 m. We then calculate
the root mean square (RMS) of the difference between $\sigma_\infty$ from this average over a 2 m sliding window, roughly equivalent
to the 10 ns window over which the return power is averaged. The resulting plots for $\sigma_\infty$ and $\mathrm{RMS}(\sigma_\infty - \sigma_{avg})$ are shown in
Fig. 2. Using the RMS of the conductivity instead of the conductivity itself is especially advantageous around 750 to 1000 m,
where the average conductivity drops. This drop should have a minor effect on reflectivity, and is likely to be spurious (Wolff
et al., 1995). It also takes into account that a rapid decrease in conductivity may cause a radio echo just as well as an increase.

The resulting correlation between radio return power and $\mathrm{RMS}(\sigma_\infty - \sigma_{avg})$ for different values of $\Delta T$ and $n$ is shown in
Fig. 3. It shows a clear maximum at a value of $n = 1.778$.

Using this result, we plot the radio return power as a function of reflector depth along with ice conductivity. The result
(Fig. 4) shows a good correlation between the two. Most jumps in conductivity are matched with a radio echo, though there
are exceptions, most notably at 520m. There are a few radio echos that do not seem to have a corresponding feature in the
conductivity data, for example at 230 m. Similar inconsistencies between radio reflections and ice conductivity have also been
noted by other measurements Eisen et al. (2003).

If we repeat this process with the combined measurements from all three runs, the result is very similar to the one obtained
from using just the measurement at the GISP2 hole, with the maximum correlation for an index of refraction of $n = 1.774$.
Superimposing the radio return power and the ice conductivity (Fig. 5) shows that the correlation holds down to larger depths.
As the deeper measurements were taken at a considerable distance from the GISP2 hole, there may be a change in the depths of
some reflectors. The good fit between radio echos and $\sigma_\infty$ suggests that this change is small, if present at all. Still, it represents
a potentially significant and difficult to estimate uncertainty on the index of refraction measurement, which is why we prefer
the measurement taken at the GISP2 hole itself. But it demonstrates that this index of refraction measurement can be extended
to greater depths relatively easily, if desired.





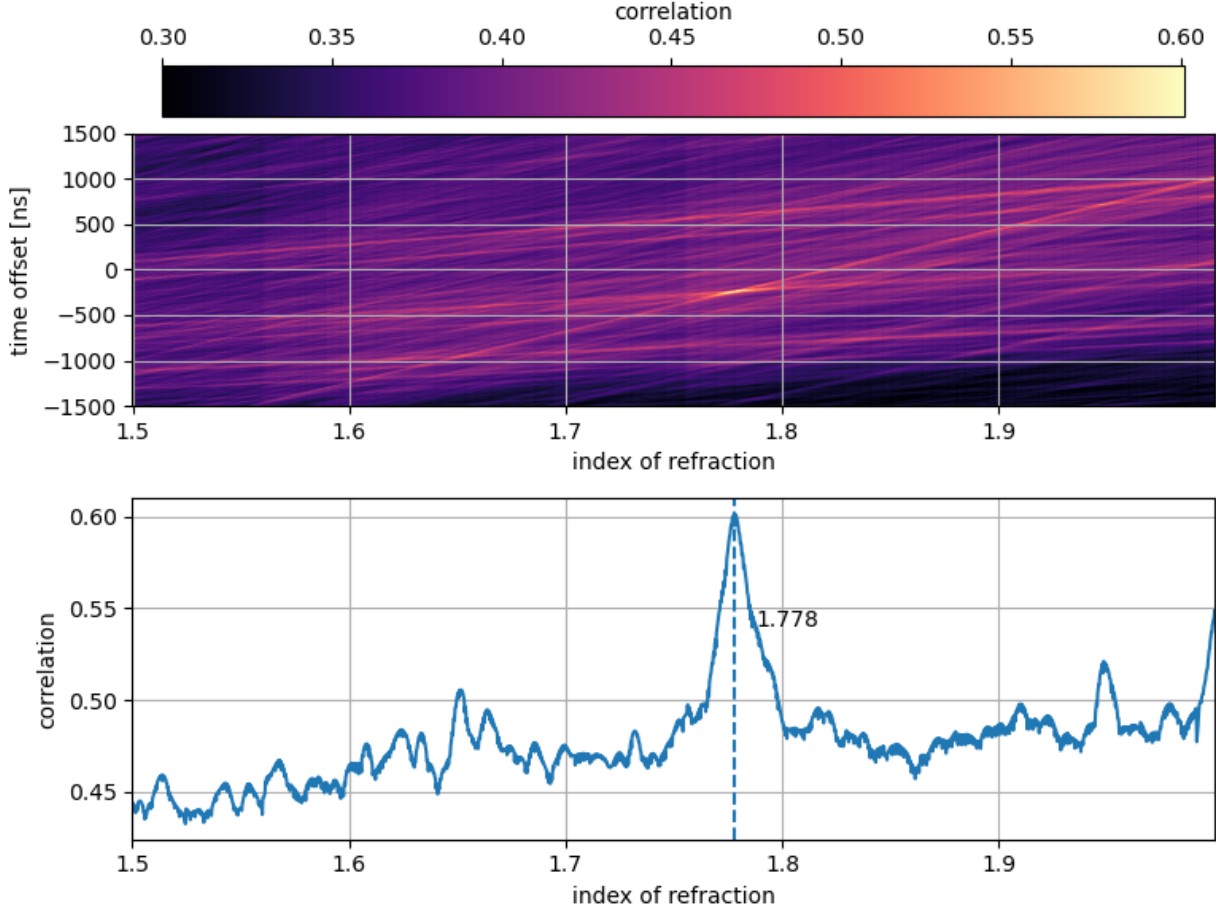

**Figure 3.** Top: Correlations between radio return power and $\mathrm{RMS}(\sigma_\infty - \sigma_{avg})$ for a given combination of index of refraction and time offset values. Bottom: Maximum correlation between radio return power and ice conductivity for a given index of refraction.

## 4 Uncertainty Estimation

As shown by Eq. 2, the two primary types of uncertainty we need to consider are uncertainties on the radio echo return time, and the depth of the corresponding reflective layer.

By including a global time offset as a free parameter, we are effectively only considering the time difference between reflections from different layers, so cable delays, the changing index of refraction profile in the firn and the height of the antennas relative to the 0 m mark of the GISP2 core can be ignored, as they affect the signal from each reflector equally. The waveforms for each run were recorded on a single trace with a sampling rate of 2.5 GHz, giving sub-nanosecond precision for the waveforms returning from different reflectors. The return power was integrated over a 10 ns window, which we conservatively







**Figure 4.** Radio return power as a function of the corresponding reflector depth, calculated using the reconstructed index of refraction $n$ and time offset $\Delta T$ (solid gray line), overlaid with the AC conductivity of the ice (dashed blue line).





**Figure 5.** Same plot as Fig. 4, but with the measurements near the Bally building included.





take as the uncertainty on $\Delta t$. The first and the last radio echo that can be clearly associated with a specific peak in the ice conductivity are at about 2.5 μs (195 m) and 10.2 μs (845m), respectively, resulting in a relative uncertainty of $\sigma_t = 0.1\%$.

The uncertainty on the depth of the GISP2 conductivity data is given as 2 to 3 m at 3 km (Greenland Ice Core Project, 1994).
We take this as an upper limit, though over the $\sim$650 m range in depth we are looking at, the true uncertainty is likely much smaller. The uncertainty on the matching between the GISP2 and GRIP ice cores is given as 0.5 m, except for some depths which are outside the range used in this measurement. Thus, the conservative 2 m uncertainty on the GISP2 depth scale is the dominant uncertainty, which is equal to the 2 m window over which $\mathrm{RMS}(\sigma_\infty - \sigma_{avg})$ was calculated. Over a depth range of 650 unitm, this yields a relative uncertainty of $\sigma_z = 0.3\%$.

Quadratically adding the relative uncertainties on $\Delta z$ and $\Delta t$ results in a relative uncertainty of $\sigma_n = 0.3\%$, or $\sigma_{n,abs} = 0.006$
in absolute terms.

Glacial ice has been shown to have birefringent properties, leading to a polarization dependence of the wave velocity that is larger than the uncertainty on our measurement in several places in Greenland Zeising et al. (2023); Gerber et al. (2022). We have investigated birefringence in the ice at Summit Station before Aguilar et al. (2022c), and constrained the difference in propagation time between polarizations parallel and perpendicular to the direction of ice flow to $1.6\pm3.3$ over the full thickness
of the ice sheet. Based on this, we conclude that birefringence effects are negligible for our measurement.

This is larger than the difference between our two measurements, so even without knowing the uncertainty on the measurements at the Bally building, we can say that they agree within uncertainties.

## 5  Conclusion and Outlook

We report on the observation of reflective layers in the ice sheet near Summit Station, Greenland and compare them to con-
ductivity measurements from the GISP2 and GRIP ice cores. We show that certain radio echos can be attributed to features in the ice conductivity, and use this relationship to measure the index of refraction of the bulk ice as $n = 1.778 \pm 0.006$. Though the available equipment limited our measurement to the upper $\sim$850m of the ice sheet, we show that the relation between ice conductivity and radio reflections holds to much greater depths. This would allow to easily extend this measurement and improve on its accuracy in the future. An extension of this measurement, with a wider frequency response, could, in principle,
also correlate the characteristics of the observed radar echoes with the known GISP2 ice chemistry.

*Acknowledgements.* We are thankful to the staff at Summit Station for supporting our deployment work in every way possible. Also to our colleagues from the British Antarctic Survey for embarking on the journey of building and operating the BigRAID drill for our project.

We would like to acknowledge our home institutions and funding agencies for supporting the RNO-G work; in particular the Belgian Funds for Scientific Research (FRS-FNRS and FWO) and the FWO programme for International Research Infrastructure (IRI), the National
Science Foundation (NSF Award IDs 2118315, 2112352, 211232, 2111410) and the IceCube EPSCoR Initiative (Award ID 2019597), the German research foundation (DFG, Grant NE 2031/2-1), the Helmholtz Association (Initiative and Networking Fund, W2/W3 Program), the



University of Chicago Research Computing Center, and the European Research Council under the European Unions Horizon 2020 research and innovation programme (grant agreement No 805486).

## Team List

Juan A. Aguilar[1], Patrick Allison[2], Dave Z. Besson[3], Abigail Bishop[4], Olga Botner[5], Sjoerd Bouma [6], Stijn Buitink[7], Whitmaur Castiglioni[8], Maddalena Cataldo[6], Brian A. Clark[9], Alan Coleman[5], Kenneth Couberly[3], Zachary Curtis-Ginsberg[4], Paramita Dasgupta[1], Simon de Kockere[10], Krijn D. de Vries[10], Cosmin Deaconu[8], Michael A. DuVernois[4], Anna Eimer[6], Christian Glaser[5], Allan Hallgren[5], Steffen Hallmann[11], Jordan C. Hanson[12], Bryan Hendricks[13], Jacob Henrichs[11,6], Nils Heyer[5], Christian Hornhuber[3], Kaeli Hughes[8,13], Timo Karg[11], Albrecht Karle[4], John L. Kelley[4], Michael Korntheuer[1], Marek Kowalski[11,14], Ilya Kravchenko[15], Ryan Krebs[13], Robert Lahmann[6], Paul Lehmann[6], Uzair Latif[10], Philipp Laub[6], Chao-Hsuan Liu[15] Joseph Mammo[15], Matthew J. Marsee[16], Zachary S. Meyers[11,6], Kelli Michaels[8], Katahrine Mulrey[17], Marco Muzio[13], Anna Nelles[11,6], Alexander Novikov[18], Alisa Nozdrina[3], Eric Oberla[8], Bob Oeyen[19], Ilse Plaisier[6,11], Noppadol Punsuebsay[18], Lilly Pyras[11,6], Dirk Ryckbosch[19], Felix Schlüter[1], Olaf Scholten[10,20], David Seckel[18], Mohammad F. H. Seikh[3], Daniel Smith[8], Jethro Stoffels[10], Daniel Southall[8], Karen Terveer[6], Simona Toscano[6], Delia Tosi[4], Dieder J. Van Den Broeck[10,7], Nick van Eijndhoven[10], Abigail G. Vieregg[8], Janna Z. Vischer[6], Christoph Welling[8], Dawn R. Williams[16], Stephanie Wissel[13], Robert Young[3], Adrian Zink[6]

[1] Université Libre de Bruxelles, Science Faculty CP230, B-1050 Brussels, Belgium

[2] Dept. of Physics, Center for Cosmology and AstroParticle Physics, Ohio State University, Columbus, OH 43210, USA

[3] University of Kansas, Dept. of Physics and Astronomy, Lawrence, KS 66045, USA

[4] Wisconsin IceCube Particle Astrophysics Center (WIPAC) and Dept. of Physics, University of Wisconsin-Madison, Madison, WI 53703, USA

[5] Uppsala University, Dept. of Physics and Astronomy, Uppsala, SE-752 37, Sweden

[6] Erlangen Center for Astroparticle Physics (ECAP), Friedrich-Alexander-University Erlangen-Nürnberg, 91058 Erlangen, Germany

[7] Vrije Universiteit Brussel, Astrophysical Institute, Pleinlaan 2, 1050 Brussels, Belgium

[8] Dept. of Physics, Enrico Fermi Inst., Kavli Inst. for Cosmological Physics, University of Chicago, Chicago, IL 60637, USA

[9] Department of Physics, University of Maryland, College Park, MD 20742, USA

[10] Vrije Universiteit Brussel, Dienst ELEM, B-1050 Brussels, Belgium

[11] Deutsches Elektronen-Synchrotron DESY, Platanenallee 6, 15738 Zeuthen, Germany

[12] Whittier College, Whittier, CA 90602, USA



[13] Dept. of Physics, Dept. of Astronomy & Astrophysics, Penn State University, University Park, PA 16801, USA

[14] Institut für Physik, Humboldt-Universität zu Berlin, 12489 Berlin, Germany

[15] Dept. of Physics and Astronomy, Univ. of Nebraska-Lincoln, NE, 68588, USA

[16] Dept. of Physics and Astronomy, University of Alabama, Tuscaloosa, AL 35487, USA

[17] Dept. of Astrophysics/IMAPP, Radboud University, PO Box 9010, 6500 GL, The Netherlands

[18] Dept. of Physics and Astronomy, University of Delaware, Newark, DE 19716, USA

[19] Ghent University, Dept. of Physics and Astronomy, B-9000 Gent, Belgium

[20] Kapteyn Institute, University of Groningen, Groningen, The Netherlands



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
