# Peer review of "Precision measurement of the index of refraction of deep glacial ice at radio frequencies at Summit Station, Greenland"

_EGUsphere, 2023_

## Referee Comment (RC1)

**Review: Welling and the RNO-G Collaboration; Precision measurement of the index of refraction of deep glacial ice at radio frequencies at Summit Station, Greenland**

Manuscript #: egusphere-2023-745

I was happy to be provided with the opportunity to review a manuscript that leverages measurements of ice conductivity and radio-wave englacial reflections to determine a specific index of refraction for glacial ice at the Radio Neutrino Observatory Greenland, Summit Station, on the Greenland Ice Sheet. I have limited expertise to understand the specifics of the radar hardware used, and I hope that a more qualified reviewer is able to confirm that the instrument setup was appropriate for the experiment. My suggestions to address comments below are my own personal opinion, and the authors are free to take alternative approaches so long as it is justified. Please consider my comments and suggestions below, and I apologise if I may have missed or misconstrued elements of your manuscript.

TJ Young (University of St Andrews)
31 July 2023

**General comments**

Overall—a succinct and clearly-written manuscript and I have several comments, of which all are relatively minor.

**(a) Further clarity on radio wave properties**

Please provide the signal frequencies/wavelengths generated. Even within the radio frequency band, most of the equations that the manuscript results employ rely on assumptions, which may or may not be negligible depending on the nature of the radio wave properties used by the instrument. Frequency affects the permittivity variations, which in turn relates to the index of refraction. Hence, for example, your assumption of negligible polarisation dependence on your calculations (L175) can only be justified if the frequencies used are the same or within a threshold to those used in Aguilar et al. (2022c). Separately, though not affecting the actual calculations, frequency will also affect the strength of observed conductivity-induced englacial reflections (Fujita & Mae 1994).

**(b) Explicit statement needed that this method assumes additional invariance in several parameters**

Given that the study calculates a bulk index of refraction, there is then an implicit assumption of a constant permittivity in and density of ice, following Looyenga (1965). At this point, the framework needs to assume that reflections detected by the radio wave is from abrupt contrasts in conductivity and not from permittivity (Fujita & Mae 1994). Note also that englacial reflections can also be caused by changes in density and crystal orientation fabric.

Additionally, the speed of the radar wave when travelling through ice also is dependent on density, temperature, and crystal orientation fabric (the last of which is already mentioned in (a)). Density is addressed for the most part (I think) by absorbing it into the $\Delta T$ free parameter (via the offset in time by firn properties). Within the depth ranges that you are working at, the effects of temperature should be negligible. However, if you extend your analyses to shallower or deeper sections of the ice column, it should be taken into consideration, or at least shown to (still) be negligible.

I would then suggest to make explicit that the method used in this manuscript assumes invariance in these factors (permittivity, temperature) for both radio-wave transmission and reflection. A good citation for this would be Fujita et al. (2000).

**(c) Consideration of the "echo-free zone"**

The manuscript presents confident claims that the method can be applied "to greater depths relatively easily". There is however a common occurrence of an 'echo-free zone' (e.g. Drews et al. 2009) in which, for reasons still largely unknown, radars are unable to consistently receive coherent englacial reflections. I would suggest to take this caveat into consideration.

**(d) Suggestions to consider employing ice-penetrating data to strengthen the argument**

The suggestion that the differences between reflectors at the two different radio-echo sounding sites, as well as their comparison to the GISP2 borehole, can perhaps be verified through visual interpretation of ice-penetrating radar profiles done around the site, depending on exactly where and which directions the measurements were taken relative to the surrounding landmarks. See if the radargrams provided in Jacobel & Hodge (1995) may help towards this suggestion. There may be other radargrams that exist at resolutions that may be too coarse for beneficial use (e.g. the CReSIS datasets).

**(e) Formatting**

I have not commented on issues with formatting, but the authors should give the manuscript a thorough proofreading to correct typos and formatting issues, especially with citations (`\citep` v. `\citet` in LATEX).

**Specific comments (by line L)**

| | |
|---|---|
| 10 | "Cosmic rays have been of interest to physicists for over a hundred years" ← why? The journal has a wide readership so providing more basis will strengthen your motivation for research. |
| 16-17 | Spell out Eev (exa-electronvolts?) It would also perhaps help those not familiar with neutrino physics if you can also mention that the energy produced by cosmic neutrinos can be in this range. Does the detection of high-energy particles then scale proportionally with size, such that the detectors have to be proportionally large? |
| 70 | How "near" was the first measurements to the GISP2 borehole? |
| 86-92 | To just check, were the same antennas used in these two additional measurements and were they also positioned 106m apart? Was the antenna plane aligned in the same direction as the first measurement? |
| 86 | Which direction were the measurements (here I am assuming the 2nd and 3rd) taken from the GISP2 hole? |
| 100 | Note also, though, that there is considerable variation between conductivity measurements between adjacent ice cores (Wolff et al. 2005, Gautier et al. 2016). |
| 105 | Suggest taking this first sentence out, this is your opinion. |
| Eq. 4 | From this equation it is now evident that $z_0$ is the vertical distance that also takes into account planar distance, and $z$ is the vertical distance without this deviation. However Eq. 3 implies the opposite (that $z_0$ is instead the vertical distance with no horizontal deviation). |
| 124 | Quantify how "negligible" is this effect. |

| | |
|---|---|
| 125 | A statement that firn variations around the GISP2 site is negligible would support your case here, if there is a study that exists. |
| Fig. 3 | I'm not quite sure where the values for the bottom plot are coming from. Are these max correlation values for a given index of refraction regardless of which time offset they represent, or for the specific time offset that gives the maximum correlation value of 1.778? |
| 157 | Same argument as my comment for L125: this statement is true only if there is no areal variation in firn density if you want to include measurements taken from different locations. |
| Figs. 4, 5 | This is my personal opinion, but I can get all the information that I need from Figure 5, which renders Figure 4 unnecessary. |
| 165 | Please give the citation that provides the 0.5m uncertainty measurement. |
| 168 | 650 m |
| 170 | I am not certain that simply adding the uncertainty measurements on $\Delta z$ and $\Delta t$ is the correct way to produce a corresponding uncertainty for $n$ especially given their placement in Eq. 1. |
| 174 | Give units for $1.6 \pm 3.3$ |
| 176 | As far as I understand, there are three measurements, one taken "near the GISP2 borehole" (L70-71) and two taken "550m away from the GISP2 borehole" (L86) – which of these are the "two measurements" that you are referring to? |
| 183 | "… and radio reflections *should* hold…" ← given that you have not shown this evidence yet. |

**References**

Drews, R., Eisen, O., Weikusat, I., Kipfstuhl, S., Lambrecht, A., Steinhage, D., ... & Miller, H. (2009). Layer disturbances and the radio-echo free zone in ice sheets. *The Cryosphere*, *3*(2), 195-203.

Fujita, S., & Mae, S. (1994). Causes and nature of ice-sheet radio-echo internal reflections estimated from the dielectric properties of ice. *Annals of Glaciology*, *20*, 80-86.

Gautier, E., Savarino, J., Erbland, J., Lanciki, A., & Possenti, P. (2016). Variability of sulfate signal in ice core records based on five replicate cores. *Climate of the Past*, *12*(1), 103-113.

Jacobel, R. W., & Hodge, S. M. (1995). Radar internal layers from the Greenland summit. *Geophysical Research Letters*, *22*(5), 587-590.

Looyenga, H. (1965). Dielectric constants of heterogeneous mixtures. *Physica*, *31*(3), 401-406.

Wolff, E. W., Cook, E., Barnes, P. R., & Mulvaney, R. (2005). Signal variability in replicate ice cores. *Journal of Glaciology*, *51*(174), 462-468.

---

## Referee Comment (RC3)

**Final comments to review: Welling and the RNO-G Collaboration; Precision measurement of the index of refraction of deep glacial ice at radio frequencies at Summit Station, Greenland**

Manuscript #: egusphere-2023-745

Thank you to Christoph Welling and co-authors for responding point-by-point to my initial review to your manuscript.

I am posting a response with final comments ahead of the release of a revised version of the manuscript, as I am shortly heading to Antarctica on fieldwork with limited to no internet access. I am happy for the manuscript to be published once the authors address these final points (all minor) below. In the mean time, in the spirit of progress I am assuming that the revised manuscript accurately reflects the changes mentioned in your 13 August response.

TJ Young (University of St Andrews)
23 October 2023

**General comments**

Separated into the same sections as the initial review, I have some last comments as well as several requests as you prepare your manuscript towards publication:

**(a) Further clarity on radio wave properties**

Thank you for clarifying that your antenna setup is the same as Aguilar et al. (2022c). Please make this clear and explicit in your manuscript, perhaps in the Methods section. I would also recommend that you include a statement somewhere in the manuscript that frequency will affect the strength of the observed conductivity-induced englacial reflections (Fujita & Mae 1994), which is a caveat that dictates the optimum range of frequencies that could be used to conduct a similar experiment in the future.

**(b) Explicit statement needed that this method assumes additional invariance in several parameters**

I am glad that you have added a brief paragraph discussing the assumption of invariance in density, permittivity, temperature, and crystal orientation fabric. I hope this paragraph also addresses that reflections detected by the radio wave are assumed to arise from abrupt contrasts from conductivity and not from permittivity (Fujita & Mae 1994).

**(c) Consideration of the "echo-free zone"**

I agree that given the data you present, the method could potentially be applied to data to 1700 m. It was not clear until L182 that you had limited your measurements to the upper ~850 m in ice column. This perhaps should be stated much earlier in the manuscript, such as in the beginning of the results section or in the Methods section.

**(d) Suggestions to consider employing ice-penetrating data to strengthen the argument**

I'm glad that you found the radio-echo sounding transects from Jacobel & Hodge (1995) helpful.

**Specific comments (by line L)**

86-92          Thanks for clarifying that your antennas were the same and positioned closer than the

setup at the GISP2 hole. I would recommend stating this in the manuscript even though they were not used for the refraction measurements, as you still show the data to lower depths and use these results to suggest that the method can hold over deeper domains.

176      Thanks for clarifying the two measurements of $n$—I'd recommend making this explicit in the manuscript at or around this Line.

**References**

Fujita, S., & Mae, S. (1994). Causes and nature of ice-sheet radio-echo internal reflections estimated from the dielectric properties of ice. *Annals of Glaciology*, *20*, 80-86.

Jacobel, R. W., & Hodge, S. M. (1995). Radar internal layers from the Greenland summit. *Geophysical Research Letters*, *22*(5), 587-590.

---

## Author Comment (AC1)

Review: Welling and the RNO-G Collaboration; Precision
measurement of the index of refraction of deep glacial ice at radio
frequencies at Summit Station, Greenland
Manuscript #: egusphere-2023-745

I was happy to be provided with the opportunity to review a manuscript that leverages
measurements of ice conductivity and radio-wave englacial reflections to determine a
specific index of refraction for glacial ice at the Radio Neutrino Observatory Greenland,
Summit Station, on the Greenland Ice Sheet. I have limited expertise to understand the
specifics of the radar hardware used, and I hope that a more qualified reviewer is able to
confirm that the instrument setup was appropriate for the experiment. My suggestions to
address comments below are my own personal opinion, and the authors are free to take
alternative approaches so long as it is justified. Please consider my comments and
suggestions below, and I apologise if I may have missed or misconstrued elements of your
manuscript.
TJ Young (University of St Andrews)

Thank you for your review. Your comments were very helpful, and we followed your advice
on most things. Point by point answers are below.

31 July 2023
General comments
Overall—a succinct and clearly-written manuscript and I have several comments, of which all
are relatively minor.
(a) Further clarity on radio wave properties
Please provide the signal frequencies/wavelengths generated. Even within the radio
frequency band, most of the equations that the manuscript results employ rely on
assumptions, which may or may not be negligible depending on the nature of the radio wave
properties used by the instrument. Frequency affects the permittivity variations, which in turn
relates to the index of refraction. Hence, for example, your assumption of negligible
polarisation dependence on your calculations (L175) can only be justified if the frequencies
used are the same or within a threshold to those used in Aguilar et al. (2022c). Separately,
though not affecting the actual calculations, frequency will also affect the strength of
observed conductivity-induced englacial reflections (Fujita & Mae 1994).

The lower end of the frequency band is set by the 145MHz highpass filter mentioned in the
description of the measurement setup. We added the information that the upper end of the
band is at about 500MHz, set by the response of the horn antenna.
This is the same band as Aguilar et al. (2022c), so assuming a negligible polarization
dependence is justified.

(b) Explicit statement needed that this method assumes additional invariance in several
parameters Given that the study calculates a bulk index of refraction, there is then an implicit
assumption of a constant permittivity in and density of ice, following Looyenga (1965). At this
point, the framework needs to assume that reflections detected by the radio wave is from
abrupt contrasts in conductivity and not from permittivity (Fujita & Mae 1994). Note also that
englacial reflections can also be caused by changes in density and crystal orientation fabric.
Additionally, the speed of the radar wave when travelling through ice also is dependent on

density, temperature, and crystal orientation fabric (the last of which is already mentioned in (a)). Density is addressed for the most part (I think) by absorbing it into the ΔT free parameter (via the offset in time by firn properties). Within the depth ranges that you are working at, the effects of temperature should be negligible. However, if you extend your analyses to shallower or deeper sections of the ice column, it should be taken into consideration, or at least shown to (still) be negligible.

I would then suggest to make explicit that the method used in this manuscript assumes invariance in these factors (permittivity, temperature) for both radio-wave transmission and reflection. A good citation for this would be Fujita et al. (2000).

We added a brief paragraph at the end of the discussion about uncertainties discussing this. Temperature profiling of the GISP2 borehole shows a constant temperature within 1°C for the upper 2km, so the effect of temperature variations can be safely assumed to be negligible for the range of depths used in this measurement.
As the vast majority of neutrino interactions detected by RNO-G are expected to occur in the upper ~1.5km of the ice sheet, this assumption also holds for the detection volume of RNO-G.

(c) Consideration of the "echo-free zone"
The manuscript presents confident claims that the method can be applied "to greater depths relatively easily". There is however a common occurrence of an 'echo-free zone' (e.g. Drews et al. 2009) in which, for reasons still largely unknown, radars are unable to consistently receive coherent englacial reflections. I would suggest to take this caveat into consideration.

We show in Fig. 5 that the correlation between radar echos and changes in ice conductivity holds to at least 1700m, which would allow us to at least double the depth range used for the measurement. That is what we meant by saying that this measurement can easily be extended.

(d) Suggestions to consider employing ice-penetrating data to strengthen the argument
The suggestion that the differences between reflectors at the two different radio-echo sounding sites, as well as their comparison to the GISP2 borehole, can perhaps be verified through visual interpretation of ice-penetrating radar profiles done around the site, depending on exactly where and which directions the measurements were taken relative to the surrounding landmarks. See if the radargrams provided in Jacobel & Hodge (1995) may help towards this suggestion. There may be other radargrams that exist at resolutions that may be too coarse for beneficial use (e.g. the CReSIS datasets).

Thanks you for this suggestion. Jacobel & Hodge explicitly mention that (except for the deepest 300-400m), the internal reflective layers are continuous between GISP2 and GRIP. We now mention this in the paper when we justify using the GRIP conductivity data for GISP2.

(e) Formatting
I have not commented on issues with formatting, but the authors should give the manuscript a thorough proofreading to correct typos and formatting issues, especially with citations (\citep v.

\citet in LATEX).

Specific comments (by line L)

10 "Cosmic rays have been of interest to physicists for over a hundred years" ß why? The journal has a wide readership so providing more basis will strengthen your motivation for research.
16-17 Spell out Eev (exa-electronvolts?) It would also perhaps help those not familiar with neutrino physics if you can also mention that the energy produced by cosmic neutrinos can be in this range. Does the detection of high-energy particles then scale proportionally with size, such that the detectors have to be proportionally large?

This is a good point, we expanded the introduction a bit to give some more background on UHE cosmic rays and neutrinos.

70 How "near" was the first measurements to the GISP2 borehole?

This is described a little bit later, where the setup is described as having the antennas 102 meters apart with the GISP2 hole in the middle. We rephrased this a bit to make it clear that this means they are 51m to each side of the hole.

86-92 To just check, were the same antennas used in these two additional measurements and were they also positioned 106m apart? Was the antenna plane aligned in the same direction as the first measurement?

These were the same antennas. They were closer together than the 102m at the GISP2 hole, but for a depth of 550m the change in propagation length due to horizontal separation between the antennas is small (about 2m). Please note that these measurements were not used for the index of refraction measurement, but just to demonstrate that the correlation between radio echos and conductivity holds to larger depths.
The antenna alignment was the same as in the first measurement.

86 Which direction were the measurements (here I am assuming the 2nd and 3rd) taken from the GISP2 hole?

It was to  the north of the GISP2 hole. We added that information to the paper.

100 Note also, though, that there is considerable variation between conductivity measurements between adjacent ice cores (Wolff et al. 2005, Gautier et al. 2016).

This was our motivation for not using the additional measurements to determine the index of refraction, as we could not guarantee that the reflective layers were still at the same depths further from the borehole.

105 Suggest taking this first sentence out, this is your opinion.

Done

Eq. 4 From this equation it is now evident that z0 is the vertical distance that also takes into account planar distance, and z is the vertical distance without this deviation. However Eq. 3 implies the opposite (that z0 is instead the vertical distance with no horizontal deviation).

The notation is indeed a bit inconsistent here. We fixed it

124 Quantify how "negligible" is this effect.

The difference in signal travel time when including raytracing is less than 1ns. We added this information.

125 A statement that firn variations around the GISP2 site is negligible would support your case here, if there is a study that exists.
157 Same argument as my comment for L125: this statement is true only if there is no areal variation in firn density if you want to include measurements taken from different locations.

The time offsets between the different measurements are corrected for (described around line 100). So if a change in the firn properties between the different sites caused a time delay, this would be corrected here as well. Therefore we do not need to assume a uniform firn.

Fig. 3 I'm not quite sure where the values for the bottom plot are coming from. Are these max correlation values for a given index of refraction regardless of which time offset they represent, or for the specific time offset that gives the maximum correlation value of 1.778?

Yes, the time offset is left to vary for each value of n. We added this information to the figure description.

Figs. 4, 5 This is my personal opinion, but I can get all the information that I need from Figure 5, which renders Figure 4 unnecessary.

We think that Figure 4 makes it easier to see details and is less cluttered, so we would prefer to keep it.

165 Please give the citation that provides the 0.5m uncertainty measurement.

The 0.5m uncertainty was given in one of the publications we already cited for the depth correction between GRIP and GISP2. We repeated the citation here again.

170 I am not certain that simply adding the uncertainty measurements on Δz and Δt is the correct way to produce a corresponding uncertainty for n especially given their placement in Eq. 1.

For products and quotients, the errors can be propagated by adding the squares of the relative errors of the individual components In our case this is:
$(\sigma\_z/z)^2 = (\sigma\_t/t)^2 + (\sigma\_z/z)^2$
So our calculation is correct.

174 Give units for 1.6 ± 3.3

Done

176 As far as I understand, there are three measurements, one taken "near the GISP2 borehole" (L70-71) and two taken "550m away from the GISP2 borehole" (L86) – which of these are the "two measurements" that you are referring to?

This is referring to the two measurements of n, one using only the data taken at GISP2, and the other one also including the data from near the Bally building.

183 "… and radio reflections should hold…" ß given that you have not shown this evidence yet

Done

---

## Author Response (AR1)

Author's Response

We already posted point-by-point replies to the reviewers during the previous discussion phase, so we will collect these replies here again and add information on where the manuscript was edited in response.
Reviewer comments are in blue, our replies in black

Reviewer 1 (TJ Young):

1st comment:

a) Further clarity on radio wave properties
Please provide the signal frequencies/wavelengths generated. Even within the radio frequency band, most of the equations that the manuscript results employ rely on assumptions, which may or may not be negligible depending on the nature of the radio wave properties used by the instrument. Frequency affects the permittivity variations, which in turn relates to the index of refraction. Hence, for example, your assumption of negligible polarisation dependence on your calculations (L175) can only be justified if the frequencies used are the same or within a threshold to those used in Aguilar et al. (2022c). Separately, though not affecting the actual calculations, frequency will also affect the strength of observed conductivity-induced englacial reflections (Fujita & Mae 1994).

The lower end of the frequency band is set by the 145MHz highpass filter mentioned in the description of the measurement setup. We added the information that the upper end of the band is at about 500MHz, set by the response of the horn antenna.
This is the same band as Aguilar et al. (2022c), so assuming a negligible polarization dependence is justified
We changed l79 to make it clear that this setup is the same as Aguilar et al., except for the used antennas and location. We also added information on the used frequency band in l93.

(b) Explicit statement needed that this method assumes additional invariance in several parameters Given that the study calculates a bulk index of refraction, there is then an implicit assumption of a constant permittivity in and density of ice, following Looyenga (1965). At this point, the framework needs to assume that reflections detected by the radio wave is from abrupt contrasts in conductivity and not from permittivity (Fujita & Mae 1994). Note also that englacial reflections can also be caused by changes in density and crystal orientation fabric. Additionally, the speed of the radar wave when travelling through ice also is dependent on density, temperature, and crystal orientation fabric (the last of which is already mentioned in (a)). Density is addressed for the most part (I think) by absorbing it into the ΔT free parameter (via the offset in time by firn properties). Within the depth ranges that you are working at, the effects of temperature should be negligible. However, if you extend your analyses to shallower or deeper sections of the ice
column, it should be taken into consideration, or at least shown to (still) be negligible.

I would then suggest to make explicit that the method used in this manuscript assumes invariance in these factors (permittivity, temperature) for both radio-wave transmission and reflection. A good citation for this would be Fujita et al. (2000)

We added a brief paragraph (starting in l189) at the end of the discussion about uncertainties discussing this.
Temperature profiling of the GISP2 borehole shows a constant temperature within 1°C for the upper 2km, so the effect of temperature variations can be safely assumed to be negligible for the range of depths used in this measurement.
As the vast majority of neutrino interactions detected by RNO-G are expected to occur in the upper ~1.5km of the ice sheet, this assumption also holds for the detection volume of RNO-G.

(c) Consideration of the "echo-free zone"
The manuscript presents confident claims that the method can be applied "to greater depths relatively easily". There is however a common occurrence of an 'echo-free zone' (e.g. Drews et al. 2009) in which, for reasons still largely unknown, radars are unable to consistently receive coherent englacial reflections. I would suggest to take this caveat into consideration.

We show in Fig. 5 that the correlation between radar echos and changes in ice conductivity holds to at least 1700m, which would allow us to at least double the depth range used for the measurement. That is what we meant by saying that this measurement can easily be Extended.

(d) Suggestions to consider employing ice-penetrating data to strengthen the argument
The suggestion that the differences between reflectors at the two different radio-echo sounding sites, as well as their comparison to the GISP2 borehole, can perhaps be verified through visual interpretation of ice-penetrating radar profiles done around the site, depending on exactly where and which directions the measurements were taken relative to the surrounding landmarks. See if the radargrams provided in Jacobel & Hodge (1995) may help towards this suggestion. There may be other radargrams that exist at resolutions that may be too coarse for beneficial use (e.g. the CReSIS datasets).

Thank you for this suggestion. Jacobel & Hodge explicitly mention that (except for the deepest 300-400m), the internal reflective layers are continuous between GISP2 and GRIP. We now mention this in the paper in l128 when we justify using the GRIP conductivity data for GISP2.

Specific comments (by line L)
10 "Cosmic rays have been of interest to physicists for over a hundred years" ß why? The journal has a wide readership so providing more basis will strengthen your motivation for research.
16-17 Spell out Eev (exa-electronvolts?) It would also perhaps help those not familiar with neutrino physics if you can also mention that the energy produced by cosmic neutrinos

can be in this range. Does the detection of high-energy particles then scale proportionally with size, such that the detectors have to be proportionally large?

This is a good point, we expanded the introduction a bit to give some more background on UHE cosmic rays and neutrinos

70 How "near" was the first measurements to the GISP2 borehole?

This is described a little bit later, where the setup is described as having the antennas 102 meters apart with the GISP2 hole in the middle. We rephrased l90 a bit to make it clear that this means they are 51m to each side of the hole.

105 Suggest taking this first sentence out, this is your opinion.

Done

Eq. 4 From this equation it is now evident that z0 is the vertical distance that also takes into account planar distance, and z is the vertical distance without this deviation. However Eq. 3 implies the opposite (that z0 is instead the vertical distance with no horizontal deviation).

The notation is indeed a bit inconsistent here. We fixed it

124 Quantify how "negligible" is this effect.

The difference in signal travel time when including raytracing is less than 1ns. We added this Information in l139

125 A statement that firn variations around the GISP2 site is negligible would support your case here, if there is a study that exists.
157 Same argument as my comment for L125: this statement is true only if there is no areal variation in firn density if you want to include measurements taken from different locations.

The time offsets between the different measurements are corrected for (described around line 100). So if a change in the firn properties between the different sites caused a time delay, this would be corrected here as well. Therefore we do not need to assume a uniform firn.

Fig. 3 I'm not quite sure where the values for the bottom plot are coming from. Are these max correlation values for a given index of refraction regardless of which time offset they represent, or for the specific time offset that gives the maximum correlation value of 1.778?

Yes, the time offset is left to vary for each value of n. We added this information to the figure Description

165 Please give the citation that provides the 0.5m uncertainty measurement.

The 0.5m uncertainty was given in one of the publications we already cited for the depth correction between GRIP and GISP2. We repeated the citation again in l184.

170 I am not certain that simply adding the uncertainty measurements on Δz and Δt is the correct way to produce a corresponding uncertainty for n especially given their placement in Eq. 1.

For products and quotients, the errors can be propagated by adding the squares of the relative errors of the individual components In our case this is:
$(\sigma\_z/z)^2 = (\sigma\_t/t)^2 + (\sigma\_z/z)^2$
So our calculation is correct.

174 Give units for 1.6 ± 3.3

Done

176 As far as I understand, there are three measurements, one taken "near the GISP2 borehole" (L70-71) and two taken "550m away from the GISP2 borehole" (L86) –
which of these are the "two measurements" that you are referring to?

This is referring to the two measurements of n, one using only the data taken at GISP2, and the other one also including the data from near the Bally building.

183 "… and radio reflections should hold…" ß given that you have not shown this evidence yet

Done

2nd comment:

(a) Further clarity on radio wave properties
Thank you for clarifying that your antenna setup is the same as Aguilar et al. (2022c). Please make this clear and explicit in your manuscript, perhaps in the Methods section. I would also recommend that you include a statement somewhere in the manuscript that frequency will affect the strength of the observed conductivity-induced englacial reflections (Fujita & Mae 1994), which is a caveat that dictates the optimum range of frequencies that could be used to conduct a similar experiment in the future.

The frequency band was chosen this is where RNO-G is most sensitive. We are happy to include statements about the effects of frequency on the measurements and to make it clearer that the setup is (almost) the same as in Aguilar et al.
We changed l79 to make it clear that the setup is (almost) the same as in Aguilar et al. and explain in l94 that other frequency ranges may be better for this measurement, but we chose the band based on the frequencies used by RNO-G.

Explicit statement needed that this method assumes additional invariance in several parameters
I am glad that you have added a brief paragraph discussing the assumption of invariance in density, permittivity, temperature, and crystal orientation fabric. I hope this paragraph also addresses that reflections detected by the radio wave are assumed to arise from abrupt contrasts from conductivity and not from permittivity (Fujita & Mae 1994)

This was already stated as an assumption, but we are happy to point it out more clearly.
We added a sentence explicitly stating this assumption in l118.

Consideration of the "echo-free zone"
I agree that given the data you present, the method could potentially be applied to data to 1700 m. It was not clear until L182 that you had limited your measurements to the upper ~850 m in ice column. This perhaps should be stated much earlier in the manuscript, such as in the beginning of the results section or in the Methods section.

We added a sentence explaining that our measurement is limited to the upper ~800m (and why we made this choice) in l110.

Specific comments (by line L)
86-92 Thanks for clarifying that your antennas were the same and positioned closer than the setup at the GISP2 hole. I would recommend stating this in the manuscript even though they were not used for the refraction measurements, as you still show the data to lower depths and use these results to suggest that the method can hold over deeper domains

We now state the distance of the antennas from the GISP2 hole in l98.
.
176 Thanks for clarifying the two measurements of n—I'd recommend making this explicit in the manuscript at or around this Line.

We rephrased that line to make it clearer what we are talking about.

Reviewer 2 (Anonymous)

Welling et al estimate bulk index of refraction (n) of glacier ice at the GISP2 location. They use existing conductivity measurements from an ice core and find best cross-correlation between

those and radar-detected internal layers, which yields the bulk index of refraction estimate for this particular site.

The paper is very focused on estimating n at this particular location, motivated by neutrino detection, and as such the results might be only relevant to the RNO-G collaboration.

To make this paper relevant to other communities it could include discussion on existing techniques on estimating n, and the values and errors that have been derived elsewhere and from different techniques. This would put the result here in some broader context and make it clear how novel this paper is. At the moment there is a claim of the estimate of n here being the most precise for Greenland at the moment (line 45) but no support is given to this claim.

He would like to thank you for these comments and suggestions.

It is true that this publication is very focused on the particular site of Summit Station. We provide references to other index of refraction measurements in the introduction. Unfortunately, most of those do not discuss measurement uncertainties, as the index of refraction measurement was mainly a means to another end. We could add a more thorough discussion on differences between measurements by different groups, though these will likely reflect more on the variability between different locations than the measurements themselves.

We added a reference to Eisen et al. 2003 in l61, which contains a short overview of different permittivity/index of refraction measurements, which differ roughly at the percent level from each other. We also removed the claim that our measurement is the most precise for Greenland.

As far as I can tell, the approach for estimating n does not differ from that of Winter et al. If that is the case. If that is not the case, and I apologize if I missed something, it would be good to highlight the improvements/differences.

Uncertainty - supposedly there is some error that comes in during the cross-correlation that comes from the assumption that peaks in conductivity change correspond to radar-detected internal layers. This often holds, but sometimes it does not, potentially affecting the error estimate. Is that something you can quantify?

The method used in this publication is very resilient to radar reflections from sources other than changes in conductivity or changes in conductivity not leading to radio echos. These cases will result in a smaller maximum correlation compared to the other values for n, but does not affect the position of the correlation maximum. While we cannot quantify this effect, these cross correlation methods are very resilient to spurious correlations from noise, as long as there is still a clear maximum identifiable.

There is no discussion of the location of how the location of the firn/ice boundary was assessed, past which n is assumed constant. Did you have density data available?

Density measurements of the firn at Summit Station are available from R. J. Arthern et al., Journal of Geophysical Research (Earth Surface) 118, 1257 (2013) down to 100m, and show the ice density approaching a constant value. Gow et al., Journal of Geophysical Research 102, NO. C12 (1997) put the firn/ice boundary at 75-77m based on measurements from the GISP2 borehole. This is well above the 200m depth where our measurements start.

We added a sentence addressing this in l172.

Related to this, some discussion on the assumption of constant n below firn layer seems important given the particular application of neutrino detection in mind. As stated in the introduction the accurate knowledge of n is absolutely key, and I wonder how small variation of n below the firn layer matter in this case.

The authors motivate their work by assessing the error on 1% of n, but don't discuss what is the typical error on n, indeed their result is much less than 1% away from other values (e.g. Winter et al). So I wonder if using 1% error as motivation isn't just overstating the need for more precise knowledge of n.

While the Winter et al. result is much closer than 1% to ours, the difference in other measurements is around this value. Eisen et al., Journal of Glaciology, 52, 177 (2006) for example show a discrepancy at percent level between n inferred from in situ and laboratory measurements and discuss results from other measurements, arguing that these are consistent with ~1% variation. They also assume 1% as the a priori uncertainty on n.

We added a sentence in l61 pointing out the Eisen et al. paper to motivate the 1% uncertainty on n in the example.

I don't think the authors actually compare the radar-detected internal reflections to a quantity that is equivalent to the rate of change of conductivity with depth. More on that below.

I would like to know how much this estimate of n, and in particular the estimated error, helps reduce the area of sky that needs to be monitored, as opposed to using known values of n and the respective range/errors. I think including this would make it clear whether/how relevant is this paper even to the RNO-G collaboration itself.

The way the uncertainty on n affects the neutrino direction reconstruction is as follows: The direction of the neutrino can be constrained to a contour in the shape of an ellipse, with the semi major axis $a$ given by the uncertainty on the polarization. The semi minor axis $b$ of the ellipse is given by the uncertainty on the viewing angle (i.e. the angle between the neutrino direction and the direction the radio signal is emitted in). The area of this ellipse is given by $A=pi*a*b$, so it is proportional to the viewing angle resolution. For the uncertainties of 0.4° from n and 0.5° from

the viewing angle reconstruction method, this would increase the uncertainty on the viewing angle to sigma=sqrt(0.5°^2 + 0.4°^2)=0.64°, a 28% increase to the size of the uncertainty contour on the sky. It is also worth noting that for a large subset of neutrino events (those resulting in a hadronic shower only), the viewing angle reconstruction is a lot more precise (see Figure 4 in Plaisier et al. *Eur.Phys.J.C* 83 (2023) 5, 443), making the uncertainty on n more relevant. We will add a more thorough explanation on this in the next draft.

We added a sentence in l65 pointing this out.

In line and Minor comments:

Fig 1 and 5 - It would be better to make all lines thin for better visibility of detail

We made these lines thicker to try and make the graph more readable for people with color blindness. Cryosphere seems to have rather strict guidelines on this, so maybe the editor can weigh in here?

Fig 1 - How did you determine the noise level? In the green transparent curve (attenuator) it seems that there are still peaks present at the same locations as in the red curve (no attenuator) where red is not shaded.

A good indication that the noise becomes dominant is that the integrated power approaches a constant, since the radio echo is expected to keep falling off. This does not necessarily mean that there can be no echos strong enough to be detectable over the noise after that, but peaks due to noise become more likely.

45 - do you mean precise or accurate? How did you assess that? There is no discussion/overview of existing techniques and results and uncertainties.

As this is mostly about systematic uncertainties, it should be "accurate". We were a bit sloppy here. But we removed this sentence anyway.

60-64 - I don't understand this part, can you give a range for how much the area of the error ellipse increases for 1% error on n, instead of "significantly"?

We answered this in the text above.

130-136 - I don't see how the procedure described here, taking a difference between raw and smoothed signal and calculating a rms over some window is equivalent to taking a derivative. I might have missed something in the text (providing an actual formula would be much clearer) but it would be to clarify how it is that the authors are actually comparing differences of conductivity rather than conductivity itself.

Taking the RMS of the conductivity around a running mean is not the same as taking the derivative, and we do not claim so. The motivation behind using it is that, if there are multiple changes of conductivity within a short distance to each other, each would produce a radio echo, which would interfere with the others, resulting in a larger echo. We therefore chose RMS as a measure of the variability of the conductivity profile over a distance roughly equivalent to one wavelength of the radio signal.

147 - past 1500m the signals seem to decorrelate

We do not know why this happens, but there are some plausible reasons: It could be interpreted as a change in the index of refraction. It is also possible (and in our opinion more likely) that the depth of these layers at the measurement site differ from those at the GISP2 borehole, due to the distance from it. This is difficult to quantify, which is why we did not use these measurements for the index of refraction measurements.

176 - what is meant by "this"?

It is referring to the uncertainty on n. Unfortunately, this got messed up when we added the paragraph about birefringence during the editor review. Thank you for pointing this out, we will fix it

184 - What is meant by "this measurement?"

It is referring to the measurement of radio echos from ice layers, not the measurement of n itself. We rephrased this to make it clear.

Fig 4 - the dashing obscures detail, better make blue line solid too, same for Fig 2a and orange dashed line - make it thin and solid

The same as Fig1 and 5. Maybe the editor can weigh in on if making these lines solid still keeps it readable enough for colorblindness?

---

## Author Response (AR2)

Dear Editor,

Thank you for your review. We think that your suggestion to change the scope to a Brief Communication is a good idea and would like to continue along that path. As such, we have shortened the paper and edited it to conform with the Brief Communication format. Though this required a lot of changes in the text to remain coherent, we mainly reduced the detail of some explanations where we thought it was appropriate.

To answer the minor comments:

- We got rid of most of the abbreviations in the manuscript
- Radar surveys around Summit, such as Jacobel and Hodge (1995), suggest the layers are horizontal, or very close to horizontal. Since the radio echo measurements were performed close to the GISP2 hole (51m to each side) the reflector depths should match the GISP2 data very closely. For the measurements that we performed at the Bally building, the tilt may become an issue, which is why we ultimately decided not to use these measurements.
- The uncertainties on the matching between GISP2 and GRIP depths are given as 0.5m, which is stated in the manuscript. However, this is negligible compared to the uncertainty on the GISP2 depths (2-3m).

There is a request we have about the authorship:

We originally submitted this manuscript as "The RNO-G collaboration", and were then asked to change it to "Christoph Welling and the RNO-G Collaboration", so that the corresponding author is listed explicitly. This is an unusual format in physics publications and technically violates the publication guidelines of our collaboration. We realize that this is an unusual request, but since similar exceptions have been made for the IceCube collaboration in the past, we would kindly ask if we could still name "The RNO-G Collaboration" as the author.

Thank you
Christoph Welling for the RNO-G Collaboration

---

## Author Response (AR3)

Dear Editor,

thank you for your positive review. We appreciate your comment about the missing reflections, but decided to leave the wording as is. While we agree that these are nothing to worry about, they are a discrepancy from the model we are using for our measurement, so this wording still feels appropriate. Therefore, the only change to the previous version is changing the author name to "The RNO-G collaboration".

We also apologize for the delayed response.

Thank you

Christoph Welling for the RNO-G collaboration